# *MAC* Family Transcription Factors Enhance the Tolerance of Mycelia to Heat Stress and Promote the Primordial Formation Rate of *Pleurotus ostreatus*

**DOI:** 10.3390/jof10010013

**Published:** 2023-12-27

**Authors:** Kexing Yan, Lifeng Guo, Benfeng Zhang, Mingchang Chang, Junlong Meng, Bing Deng, Jingyu Liu, Ludan Hou

**Affiliations:** 1College of Food Science and Engineering, Shanxi Agricultural University, Jinzhong 030801, China; yan15934475351@163.com (K.Y.); m17836509709@163.com (L.G.); zbf18434797543@163.com (B.Z.); sxndcmc@163.com (M.C.); mengjunlongseth@hotmail.com (J.M.); dengbing_sxau@163.com (B.D.); 2Shanxi Research Center for Engineering Technology of Edible Fungi, Jinzhong 030801, China; 3Shanxi Key Laboratory of Edible Fungi for Loess Plateau, Jinzhong 030801, China

**Keywords:** *Oyster mushroom*, gene silencing, gene overexpression, stress tolerance, fruiting body

## Abstract

*Pleurotus ostreatus* is a typical tetrapolar heterologous edible mushroom, and its growth and development regulatory mechanism has become a research hotspot in recent years. The MAC1 protein is a transcription factor that perceives copper and can regulate the expression of multiple genes, thereby affecting the growth and development of organisms. However, its function in edible mushrooms is still unknown. In this study, two transcription factor genes, *PoMCA1a* and *PoMAC1b,* were identified. Afterwards, *PoMAC1* overexpression (OE) and RNA interference (RNAi) strains were constructed to further explore gene function. The results showed that the *PoMAC1* mutation had no significant effect on the growth rate of mycelia. Further research has shown that OE-*PoMAC1a* strains and RNAi-*PoMAC1b* strains exhibit strong tolerance under 32 °C heat stress. However, under 40 °C heat stress, the OE of *PoMAC1a* and *PoMAC1b* promoted the recovery of mycelial growth after heat stress. Second, the OE of *PoMAC1a* can promote the rapid formation of primordia and shorten the cultivation cycle. In summary, this study indicated that there are functional differences between *PoMAC1a* and *PoMAC1b* under different heat stresses during the vegetative growth stage, and *PoMAC1a* has a positive regulatory effect on the formation of primordia during the reproductive growth stage.

## 1. Introduction

*Pleurotus ostreatus* is rich in various active ingredients and has multiple functions, such as immune regulation [1], hypoglycemia, hypolipidemia [2], antitumor [3], antioxidant [4], antibacterial [5], and anti-inflammatory [6]. It is one of the most widely cultivated edible mushrooms in the world [7]. The growth and development of edible mushrooms can be divided into the vegetative growth stage (mycelial growth stage) and reproductive growth stage (mushroom emergence stage) [8]. During the vegetative growth stage of mushrooms, they are most likely to be affected by high-temperature environments [9,10]. Previous studies have shown that high-temperature-treated *P. ostreatus* can induce cell apoptosis, and cell death can be inhibited by adding chemicals or ROS scavengers that can block the mitochondria-induced apoptosis process [11]. Subsequently, many studies have explored the mechanism of the heat stress response in *P. ostreatus* through proteomics and transcriptomics. For example, heat stress promotes the degradation of unsaturated fatty acids and nucleotides in *P. ostreatus*, increases the content of amino acids and vitamins, and accelerates glycolysis and the tricarboxylic acid cycle [12]. Furthermore, with the mature application of genetic transformation technology in edible mushrooms, the functions of many genes and transcription factors have been reported. For example, interference with the phenylalanine ammonia lyase (*PAL*) gene enhances the tolerance of mycelia to high temperature in *P. ostreatus* [13]. Overexpression of the methionine sulfoxide reductase gene enhances the stress tolerance of *P. ostreatus* [14]. Many genes and transcription factors have also been proven to play an important role in the emergence stage of mushrooms [15,16]. For example, the *PDD1* transcription factor positively regulates the emergence of *Flammulina filiformis* and can increase their yield [17]. The transcription factor *LFC1* negatively regulates the growth, development, and yield of *F. filiformis* [18]. Interfering with the *PAL* gene can delay the formation of mushroom primordia [13]. However, compared with animals and plants, there is currently a lack of basic research on edible mushrooms, and the response mechanism to heat stress in *P. ostreatus* is still not perfect.

Many enzymes require metal ions as cofactors to function, and common metal enzyme cofactors include iron, zinc, and copper [19]. The copper sensing transcription factor MAC1 is mainly involved in regulating the absorption of Cu ions in yeast cells. The MAC1 protein contains DNA-binding domains and activation domains [20]. There is a highly conserved copper fist structure in its DNA-binding domain, which plays a crucial role in binding to the copper ion-responsive elements of downstream gene promoters [21]. There are one or two cysteine-rich motifs (REPs) in the activation domain of MAC1, which are involved in the binding of copper ions. In *Saccharomyces cerevisiae*, there is also a copper-sensing transcription factor, ACE1, which has the same copper fist structure and REP motif as MAC1 [22]. In *S. cerevisiae*, *MAC1* and *ACE1* are two opposite transcription regulatory factors that strictly control the level of free Cu^+^. When copper is excessive, ACE1 activates the detoxification and storage genes of copper. Therefore, the Cu^+^ levels detected by ACE1 set an upper limit for free copper in yeast cells. In contrast, when copper levels are low, MAC1 activates the copper uptake gene, so the Cu^+^ level perceived by MAC1 represents the lower limit of copper ions [23]. The first reported copper-sensitive transcription factor in basidiomycetes is the activating protein ACE1, expressed by Ruben et al. in *Pseudomonas aeruginosa*, which activates *CUP1* [24] expression [25]. In the filamentous fungus *Aspergillus fumigatus*, the *AfMAC1* deletion mutation resulted in slow growth and incomplete conidia, including short-chain conidia and defective melanin [26]. *AnMAC1* and its regulated Cu transporters have been shown to be necessary for the growth and conidial development of neutral *Aspergillus nidulans* during copper starvation [27]. Studies have also shown that under high-temperature stress conditions, NO activates MAC1, making cells resistant to stress, and increases SOD1 activity through CTR1-bound Cu^+^ ions [28,29]. However, little is currently known regarding the roles of MAC1 in modulating the growth, development and thermotolerance of edible mushrooms.

In this study, two *MAC1* coding genes of *P. ostreatus* were cloned. Subsequently, the functions of the two *MAC1* coding genes in response to heat stress and the growth and development of *P. ostreatus* were further explored by constructing RNA interference (RNAi) and overexpression (OE) mutants, laying the foundation for further exploration of the regulatory mechanism of MAC1.

## 2. Materials and Methods

### 2.1. Fungal Strains and Culture Conditions

The *P. ostreatus* strain CCMSSC 00389 was obtained from the China Center for Mushroom Spawn Standards and Control. The WT and mutant strains were cultured on potato dextrose agar (PDA) plates, *Escherichia coli* (DH5α) was cultured in Luria–Bertani (LB) medium containing 50 ng/mL kanamycin, and *Agrobacterium tumefaciens* (GV3101) was cultured in LB containing 50 ng/mL kanamycin and 25 ng/mL rifampicin. The PDA culture medium was purchased from Beijing Boidee Biotechnology Co., Ltd. (Beijing, China), Sensory cells purchased from TransGen Biotech (Beijing, China). Restrictive endonuclease was purchased from New England Biolabs (NEB) (Beverly, MA, USA), and DNA polymerase, reverse transcription kits, and DNA gel extraction kits were purchased from Vazyme (Nanjing, China). Primer synthesis and DNA sequencing were completed by Tsingke Biotech (Xian, China). The plasmid pBI121-EGFP was purchased from Miaoling Biology (Wuhan, China).

### 2.2. Identification, Cloning, and Sequence Analysis of PoMAC1 Genes

The *MAC1* gene sequence was obtained from the website of the National Biotechnology Information Center (https://www.ncbi.nlm.nih.gov/gene/855035) (accessed on 15 August 2014) of *S. cerevisiae* [29], and the gene ID was 855035. Then, the sequence of this gene was used to BLAST against the CCMSSC 00389 genome database [30] to identify two *PoMAC1* genes. Two PoMAC1 protein sequences are available through online websites (http://www.bioinformatics.org/sms/index.html) (accessed on 4 August 2023). A phylogenetic tree was constructed with the MEGA11 program using the neighbor joining method with a 1000 bootstrap value [31]. The MEME website (https://memesuite.org/meme/tools/meme) (accessed on 4 August 2023) was used to predict the motif of the PoMAC1 proteins. DNAMAN 6.0 software was used for multiple sequence alignment. The molecular weight, distribution of amino acids, isoelectric point (pI), and signal peptide of PoMAC1 were predicted using the online ProtParam (http://web.ExPASy.org/protparam/) (accessed on 4 August 2023). The structural domains of the PoMAC1 proteins were analyzed online (http://smart.embl-heidel-berg.de/) (accessed on 4 August 2023). The online URLs (https://swissmodel.ExPASy.org/interactive) (accessed on 4 August 2023) were used to predict the tertiary structure of the PoMAC1 proteins. The nuclear localization signals (NLS) of the PoMAC1 proteins were analyzed online (https://www.novopro.cn/tools/nls-signal-prediction.html) (accessed on 25 November 2023). All primers used in this study are shown in Appendix A.

### 2.3. Subcellular Localization of PoMAC1a and PoMAC1b

The *PoMAC1a* and *PoMAC1b* genes were cloned and inserted into the vector pBI121-EGFP through homologous recombination and then introduced into *A. tumefaciens* EHA105, with empty space as the control. Afterwards, *A. tumefaciens* EHA105 containing the target plasmid was cultured, collected and suspended in a concentrated solution (150 mM acetylsyringone, 10 mM MES monohydrate, 10 mM magnesium chloride, and pH 5.6) to achieve a final OD value of 1.0. Finally, *A. tumefaciens* was injected into tobacco leaves with good growth and cultured for 3 days. The results were observed and recorded using a confocal microscope.

### 2.4. Construction of OE and RNAi Plasmids

The *PoMAC1a* and *PoMAC1b* gene OE cassettes were constructed as follows. The original OE plasmids stored in the laboratory were digested with *Spe*I and *Psp*OMI [13]. Then, the cDNA of *PoMAC1a* and *PoMAC1b* was obtained through PCR and cloned and inserted into the vector to generate OE plasmids containing the *PoMAC1a* and *PoMAC1b* genes. Acquisition of RNAi plasmids: The original RNAi plasmids stored in the laboratory were first digested with *Spe*I and *Bgl*II, and the cloned *PoMAC1a*-sence and *PoMAC1b*-sence fragments were homologously recombined onto the plasmids. Subsequently, the obtained plasmids were digested using *Spe*I and *Psp*OMI, and the cloned *PoMAC1a*-anti and *PoMAC1b*-anti fragments were inserted to construct the interference vectors. Finally, the vectors were introduced into *A. tumefaciens* GV3101. The primers used to construct the vectors are shown in Appendix A.

### 2.5. Acquisition of Mutant Strains

First, *P. ostreatus* mycelia were inoculated onto PDA plates and cultured at 28 °C for 5 days. Then, the mycelial pellets were cut to a 5 mm diameter at the edge of the colony. Then, the mycelial pellets were placed into CYM medium and cultivated at 28 °C for 2 days. *A. tumefaciens* containing OE and RNAi plasmids were expanded and cultured, and the *A. tumefaciens* were collected in sterile tubes (50 mL capacity) by centrifugation at 4500 rpm and 4 °C for 10 min. Then, the *A. tumefaciens* cells were suspended in induction medium (IM) and incubated for 5 h. After that, the mycelial pellets were placed onto IM medium (containing *A. tumefaciens*) and cocultured at 28 °C for 5 h without shaking. Afterwards, the mycelial pellets were transferred to IM solid medium at 28 °C for 3 days and then transplanted onto CYM medium with 90 μg/mL hyg and 50 μg/mL cef. Transformants were obtained after 20–30 days [13].

### 2.6. Mushroom Production

The WT and mutant strains were cultured on PDA medium at 28 °C for 7 days. Then, the mycelia were inoculated into cottonseed shell culture medium and incubated in darkness at 25 °C for 25 days. When the mycelia were full, the culture bottles were transferred to the mushroom production box to control the production conditions. For fruiting body production, the temperature was set at 18 °C (day, 12 h at 500 lux) and at 10 °C (night, 12 h dark), and the ‘s humidity was maintained at 95% [32]. Primordia, fruiting bodies, spores, and liquid nitrogen were collected for rapid freezing, and then stored at −80 °C.

### 2.7. Heat Stress Treatment

To assess the function of *PoMAC1* in the mycelial response to heat stress, the WT, OE and RNAi strains were cultured on PDA plates at different temperatures 32 for 7 days. Afterwards, the function of *PoMAC1* was tested under 40 °C heat stress. First, *P. ostreatus* mycelia were cultured at 28 °C for 4 days, transferred to 40 °C for 24 h, and then returned to 28 °C to recovery growth for 3 days. Afterwards, the recovery growth rate of different strains of mycelia was recorded, and the growth inhibition rates of different strains were calculated [33].

### 2.8. Growth Susceptibility Assay

To assess the susceptibility of the test strains to oxidative stress, mycelial tip pellets with a 5 mm diameter were inoculated onto PDA plates supplemented with 5 mM H_2_O_2_. The diameters of the strains were measured after incubation at 28 °C for 7 days.

### 2.9. Quantitative PCR (qPCR)

To analyze the expression pattern of *PoMAC1* under different treatments, samples from different growth stages (mycelia, primordia, fruiting bodies, spores) and different periods of heat stress were collected for gene expression detection. Using the *β-tubulin* gene as the internal reference, qPCR was used to analyze the specific mRNA expression level of the *MAC1* gene. The qPCR amplification program was as follows: amplification at 95 °C for 3 min, amplification at 95 °C for 3 s, amplification at 60 °C for 32 s, 40 cycles, and amplification at 72 °C for 3 s. The relative gene expression was analyzed according to the 2^−ΔΔCT^ method. 

### 2.10. Data Analysis

SPSS 26 software was used for statistical analysis. The values are reported as the means ± SEs and were analyzed by one-way ANOVA, with a *p* value of <0.05 considered significant. GraphPad Prism 11 and Photoshop 2023 software were used for figure analysis.

## 3. Results

### 3.1. Cloning and Bioinformatics Analysis of PoMAC1a and PoMAC1b

Two MAC1 genes were obtained and identified in the genome of *P. ostreatus*, named *PoMAC1a* and *PoMAC1b*, with total cDNA lengths of 1776 bp and 1275 bp, respectively (Figure 1A). DNA sequence analysis showed that five exons of *PoMAC1a* were interrupted by four introns, and four exons of *PoMAC1b* were interrupted by three introns (Figure 1A). Bioinformatics analysis was performed on *PoAMC1a* and *PoMAC1b* sequences to determine their physicochemical properties and possible structures. The *PoMAC1a* gene encodes a peptide of 591 amino acids, with a molecular weight and pI of 63.03 kDa and 7.55, respectively. The *PoMAC1b* gene encodes a polypeptide of 424 amino acids, with a molecular weight and pI of 44.50 kDa and 9.03, respectively. To determine the evolutionary relationship between PoMAC1 identified in *P. ostreatus* and MAC1 homologous sequences reported in other fungal species, a detailed phylogenetic analysis and conservative motif comparison were conducted. The results showed that the two PoMAC1 proteins were distributed on different branches, suggesting that there might be differences in function between PoMAC1a and PoMAC1b (Figure 1B). Conservative motif analysis showed that motifs 1 and 3 existed in all MAC1 amino acid sequences. It is speculated that these two motifs may be related to the MAC1-binding site. Figure 1B shows that both PoMAC1a and PoMAC1b contain conserved motifs 1, 2, 3 and 9. However, PoMAC1a alone exhibits motifs 4, 5, 6, 8, 10, 11, and 12, and PoMAC1b alone exhibits motif 7. The amino acid comparison results showed that PoMAC1a and PoMAC1b exhibit typical Cu-sensing transcription factor characteristics, with a conserved “Cu-fist” structure at the N-terminus, which plays a crucial role in MAC1 binding to DNA. In addition, cysteine-rich motifs are widely present at the C-terminus and participate in sensing copper and stabilizing DNA binding. In Figure 2A, the Cu-fist structure is marked with a red box, while the Cys-rich motif is marked with blue. However, there is only one cysteine-rich motif, REP-Ⅰ, in the PoMAC1a, while there are two cysteine-rich motifs, REP-Ⅰ and REP-Ⅱ, in the PoMAC1b (Figure 2B). This means that different Cys-rich motifs may lead to differences in the functions of the *PoMAC1a* and *PoMAC1b* genes.

### 3.2. Subcellular Localization of PoMAC1

Transcription factors synthesized in the cytoplasm can only function by transferring to the nucleus [34]. We can verify whether PoMAC1 has nuclear localization signals through tobacco transient expression experiments. The results showed that GFP signals were present in the nucleus and cytoplasm of the positive control group, while the PoMAC1a-GFP and PoMAC1b-GFP fusion proteins were located in the nucleus (Figure 3). The research results indicated that PoMAC1 has a nuclear localization signal.

### 3.3. Acquisition of PoMAC1a and PoMAC1b Mutant Strains

To investigate the functions of *PoMAC1a* and *PoMAC1b*, OE and RNAi transformants were constructed. Figure 4A,B show the OE and RNAi plasmid maps. In the maps, the *hyg* gene was used as a selection marker to further confirm the RNAi and OE efficiency of the transformants. Then, the OE and RNAi plasmids were transferred to the mycelia of *P. ostreatus* through *Agrobacterium*-mediated genetic transformation. Figure 4C shows that the *hyg* fragment can be amplified in the mutant strains, with a size of approximately 1000 bp. Furthermore, the mutant strains were further identified by detecting gene expression levels. The results showed that compared to WT, the relative expression levels of *PoMAC1a* in OE-*PoMAC1a*-1 and OE-*PoMAC1a*-2 were approximately 2.9-fold and 2.7-fold higher, respectively. The *PoMAC1a* expression levels in RNAi-*PoMAC1a*-1 and RNAi-*PoMAC1a*-2 were reduced by 53% and 41%, respectively. The relative expression levels of *PoMAC1b* in OE-*PoMAC1b*-1 and OE-*PoMAC1b*-2 were approximately 3.2-fold and 3.6-fold higher, respectively, while the *PoMAC1b* expression levels in the RNAi strains (RNAi-*PoMAC1b*-1 and RNAi-*PoMA1b*-2) were reduced by 67% and 62%, respectively, compared to that in the WT strain (Figure 4D). Therefore, these mutant strains were selected for further research.

### 3.4. PoMAC1 Participates in the Regulation of the Mycelial Response to Different Heat Stresses

MAC1 plays an important role in various stresses. Previous studies have shown that *Saccharomyces cerevisiae* MAC1 can confer stress resistance to yeast cells under high-temperature stress [20]. In this study, we tested the function of *PoMAC1a* and *PoMAC1b* under 32 °C heat stress. The results showed that compared with the WT strain, there was no significant difference in the colony size of the RNAi-*PoMAC1a* strains grown at 32 °C for 7 days, while the colony diameter of the OE-*PoMAC1a* mutant strain significantly increased. The growth rates of the OE-*PoMAC1a*-1 and OE-*PoMAC1a*-2 strains increased by 35.83% and 23.27%, respectively, compared with the WT strain (Figure 5A,B). It is speculated that *PoMAC1a* may play a positive regulatory role under 32 °C heat stress. In the *PoMAC1b* mutant strains, OE-*PoMAC1b*-1 and OE-*PoMAC1b*-2 showed stronger heat resistance, and compared with the WT strain, their mycelial growth rates increased by 35.83% and 23.27%, respectively (Figure 5A,B). It is speculated that *PoMAC1b* may play a negative regulatory role under 32 °C heat stress. Further research has shown that *PoMAC1a* is a factor in H_2_O_2_ tolerance, as its silencing reduces growth; *PoMAC1b* is a factor for sensitivity to this oxidizing agent, as its silencing results in increased growth in the mutant. (Appendix A). In summary, it can be concluded that there are differences in the functions of *PoMAC1a* and *PoMAC1b* under heat stress at 32 °C.

To further investigate the functions of *PoMAC1a* and *PoMAC1b* in *P. ostreatus*, the expression patterns of *PoMAC1* under heat stress were detected. The results showed that there were significant differences in the expression patterns of *PoMAC1a* and *PoMAC1b* under 40 °C heat stress at different times. It can be inferred that there are also differences in functionality between *PoMAC1a* and *PoMAC1b*. In the early stage of 40 °C stress (1 h), the expression level of the *PoMAC1a* gene significantly increased and reached its peak. In contrast, with the prolongation of heat stress time, the expression level of *PoMAC1b* gradually increased and reached its peak in the later stage of stress (48 h) (Figure 6A). The growth zones of the OE-mutants of *PoMAC1a* and *PoMAC1b* were visibly more prominent than those of the control (WT) and the silenced mutants. This is evident in the peripheral zone of the colony (approximately 0.5 cm), with a finer and more exploratory mycelium showing recovery from heat stress. This exploratory growth edge is not as clearly visible in the WT or the silenced mutants, which exhibit colonies with compact edges (Figure 6B). Subsequently, the recovery growth rates of the *PoMAC1a* and *PoMAC1b* mutants were tested after heat stress at 40 °C for 24 h. The results showed that compared to the WT strain, the mycelial recovery rates of the OE-*PoMAC1a* and OE-*PoMAC1b* strains were significantly accelerated. Compared to the WT strain (80.59%), the growth inhibition rates of the OE-*PoMAC1a* and OE-*PoMAC1b* strains were reduced to 51.17% and 64.26%, respectively. In contrast, the recovery growth rates of the RNAi-*PoMAC1a* and RNAi-*PoMAC1b* strains were not significantly slower than that of the WT strains. However, the growth inhibition rates of the RNAi-*PoMAC1a* and RNAi-*PoMAC1b* strains were increased to 87.16% and 88.65%, respectively (Figure 6C). This indicates that *PoMAC1a* and *PoMAC1b* play a positive regulatory role in the response of mycelia to 40 °C heat stress.

### 3.5. PoMAC1a Positively Regulates the Formation of Primordia

To further investigate the function of the *PoMAC1a* and *PoMAC1b* genes in the growth and development of *P. ostreatus*, the expression levels of *PoMAC1a* and *PoMAC1b* in different developmental stages (mycelia, primordia, fruiting body, spores) were detected. The results showed that compared with the mycelia stage, the expression patterns of *PoMAC1a* and *PoMAC1b* were similar in both the fruiting body and spore stages, with significant downregulation of expression levels during the fruiting body stage and a significant increase in expression levels in spores. Interestingly, compared to the mycelial stage, the expression level of *PoMAC1a* in the primordia was significantly upregulated, while the expression level of *PoMAC1b* was not significantly changed (Figure 7A,B). It is speculated that *PoMAC1a* plays a key role in primordial formation. To further investigate whether *PoMAC1a* plays an important role in primordial formation, a mushroom production experiment was conducted. The results showed that compared with the WT strain, the OE-*PoMAC1a* strains promoted the primordia formation rate and shortened the growth and development cycle of fruiting bodies. In contrast, *PoMAC1a* interference prolonged the time needed for primordial formation and extended the developmental cycle of fruiting bodies. However, compared to the WT strain, the *PoMAC1b* mutant strains did not show any differences compared with the WT strain (Figure 7C,D). These results indicated that *PoMAC1a* can positively regulate the formation of primordia.

## 4. Discussion

MAC1 plays an important role in maintaining the intracellular copper ion concentration, and its function and mechanism of action in fungal yeasts have been widely studied. However, there are few reports on basidiomycetes, which have important research significance. In this study, based on a whole-genome search of *P. ostreatus*, two *MAC1* genes were identified, with *PoMAC1a* and *PoMAC1b* having 4 and 3 introns, respectively. The amino acid sequence consistency between PoMAC1a and PoMAC1b was only 22.84% (Appendix A), and in the phylogenetic tree, the genetic relationship between the two *MAC1* genes was not close, indicating that *PoMAC1a* and *PoMAC1b* are not simple gene duplications. The MAC1 protein is a copper-sensing transcription factor that participates in regulating genes essential for copper ion transport, thereby maintaining intracellular Cu concentrations [20,35]. The PoMAC1 protein has a highly conserved Cu-fist structure located in the N-terminal DNA-binding domain, and there is a cysteine-rich REP motif in its C-terminal activated domain, which is consistent with other species of MAC1 proteins. In the amino acid sequence of PoMAC1a, there is only one cysteine-rich motif, REP-I, while PoMAC1b has two different cysteine-rich motifs, REP-I and REP-II. This is similar to the study by Zhengdong Cai et al. [36]. The *A. fumigatus* copper-sensing transcription factor MAC1 homolog contains two identical Cys-rich motifs REP-Ⅰ (CXCX_3_CXCX_2_CX_2_H) at the C-terminus, while ScMAC1 in *S. cerevisiae* contains two identical Cys-rich motifs, REP-II (CXCX_4_CXCX_2_CX_2_H). In *Schizosaccharomyces pombe*, there is only one Cys-rich motif, REP-Ⅰ [36,37]. CaMAC1 in *Candida. albicans* contains two different Cys-rich motifs, REP-Ⅰ and REP-Ⅱ, at the C-terminus [38]. The REP-I of PoMAC1a and PoMAC1b in *P. ostreatus* is composed of five cysteines and one histidine, which is similar to previous research. However, the REP-II motif in PoMAC1b is composed of six cysteines and one histidine (CXCX_4_CX_1_CX_3_CX_3_CX_2_H), which is the difference between the REP-II of the MAC1 in *P. ostreatus* and the REP-II of MAC1 in other fungi [39,40]. The above results indicate that the MAC1 transcription factor is highly conserved at the N-terminus, while different changes may occur at the C-terminus, which may lead to its genes performing different functions.

MAC1 protein is involved in the growth and development of fungi and various stress responses. However, the biological function of MAC1 in edible mushrooms has not been reported. This study found that during the vegetative growth stage, there are differences in the functions of *PoMAC1a* and *PoMAC1b* in *P. ostreatus* under different heat stresses. Under mild heat stress at 32 °C, *PoMAC1a* and *PoMAC1b* exhibit opposite functions. The OE-*PoMAC1a* and RNAi-*PoMAC1b* strains promoted the growth rate of mycelia and exhibited strong heat resistance. Under extreme heat stress (40 °C), the expression patterns of *PoMAC1a* and *PoMAC1b* were similar, showing a trend of first increasing, then decreasing, and then increasing. However, *PoMAC1a* showed the highest expression in the early stage of stress (40 °C stress for 1 h), while *PoMAC1b* reached its peak in the later stage of stress (40 °C stress for 12 h). In addition, the OE of *PoMAC1a* and *PoMAC1b* can promote the recovery growth rate of mycelia after heat stress. Therefore, it is speculated that *PoMAC1a* and *PoMAC1b* may participate in regulating the response of mycelium to 40 °C heat stress at different times. Previous studies have shown that the NO produced by yeast cells under high-temperature stress activates the *MAC1*-induced *CTR1* gene, leading to an increase in intracellular copper levels. Then, Cu^+^ activates SOD1 to resist high-temperature stress [29]. Therefore, it is speculated that under 40 °C heat stress, the high expression of two *PoMAC1* genes in the mycelia of *P. ostreatus* may improve the activity of the SOD enzyme, enhance the tolerance of mycelia to reactive oxygen species, and thus promote the recovery and growth of mycelia after heat stress. In addition, the function of MAC1 has been extensively studied in other species; for example, in yeast, the functional defect mutation of *MAC1* can lead to an increase in yeast cell thermal sensitivity [20] In *Arabidopsis*, there is a copper sensing transcription factor *ACE1*, which has a highly similar structure to MAC1. Moreover, OE-*ACE1* transgenic plants have a higher survival rate under copper stress than WT plants, while increasing the activity of SOD and POD [16]. The results of this study are similar to those of previous studies, indicating that MAC1 can participate in various abiotic stress response pathways. However, it is interesting that there are significant differences in the functions of *PoMAC1a* and *PoMAC1b* in *P. ostreatus* under different stresses.

During the reproductive growth stage, *PoMAC1a* positively regulates primordial formation and shortens the developmental cycle of fruiting bodies. However, there was no significant difference between the *PoMAC1b* mutant strains and the WT strain during the reproductive stage. Previous studies have found that in *A. fumigatus*, the *AfMAC1*-deficient mutant not only significantly slows its growth rate, but also has incomplete conidia, including short chains and melanin deficiency [26]. In *S. cerevisiae*, the loss-of-function mutation of MAC1 may manifest defects in the activity of plasma membrane Cu^2+^ and Fe^3+^ reductases, slow growth, and respiratory defects [20]. Our research results are similar to those of our predecessors. These results all indicate that *MAC1* plays an important role in growth and development processes. At present, the regulatory mechanism of *PoMAC1* in the growth and development of *P. ostreatus* is still unclear. In future work, further exploration of its possible regulatory mechanism will be conducted through omics techniques.

## 5. Conclusions

In summary, this study cloned two *PoMAC1* genes from *P. ostreatus* and analyzed their protein structures. The function of *PoMAC1a* and *PoMAC1b* in the growth and development of *P. ostreatus* was explored by constructing OE and RNAi strains. During the vegetative growth stage, both *PoMAC1a* and *PoMAC1b* play a positive regulatory role under extreme heat stress (40 °C); under mild heat stress (32 °C), *PoMAC1a* plays a positive regulatory role, while *PoMAC1b* plays a negative regulatory role. During the reproductive growth stage, only *PoMAC1a* plays a positive role in the primordial formation process. This study provides a basis for exploring the role of the copper-sensitive transcription factor MAC1 in edible mushrooms.

## Figures and Tables

**Figure 1 jof-10-00013-f001:**
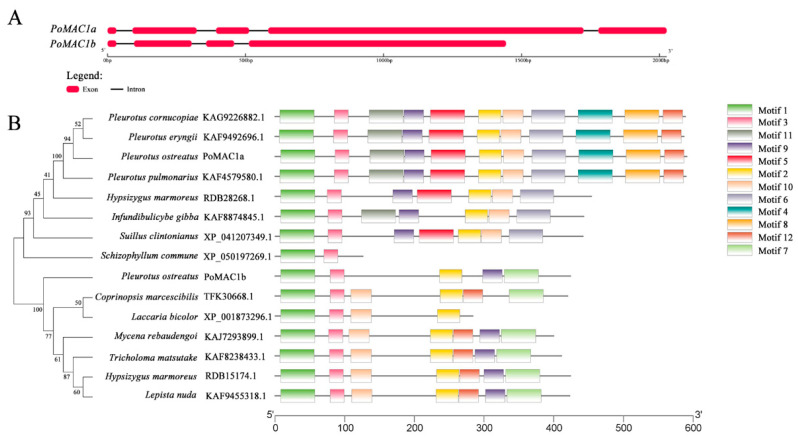
Gene structural features and relationships of fungal MAC1. (**A**) Gene structures of *PoMAC1a* and *PoMAC1b* in *P. ostreatus* CCMSSC 00389. (**B**) A neighbor-joining phylogenetic tree and motifs of 15 MAC1 protein sequences from fungal species.

**Figure 2 jof-10-00013-f002:**
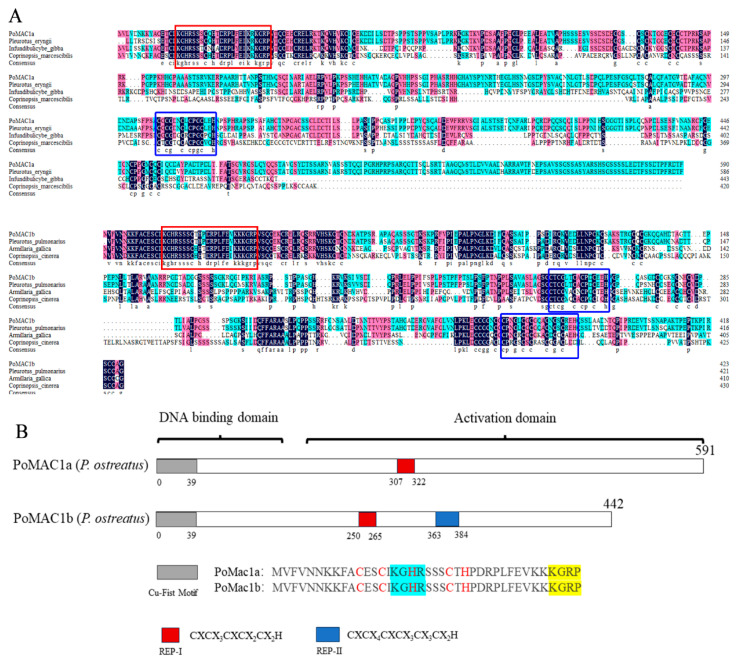
Amino acid sequence alignment and gene structure diagram of PoMAC1a and PoMAC1b. (**A**) Amino acid sequence alignment. The Cu-fist motifs of PoMAC1 are enclosed in red rectangles, while the REP-I and II motifs are enclosed in blue rectangles. (**B**) The PoMAC1 protein molecular peptide chain and the distribution of various functions. The gray area represents the Cu-fist motif, while the red and blue areas represent the REP-I and II motifs, respectively. The listed amino acid sequences are Cu-fist, REP-I, and II motifs.

**Figure 3 jof-10-00013-f003:**
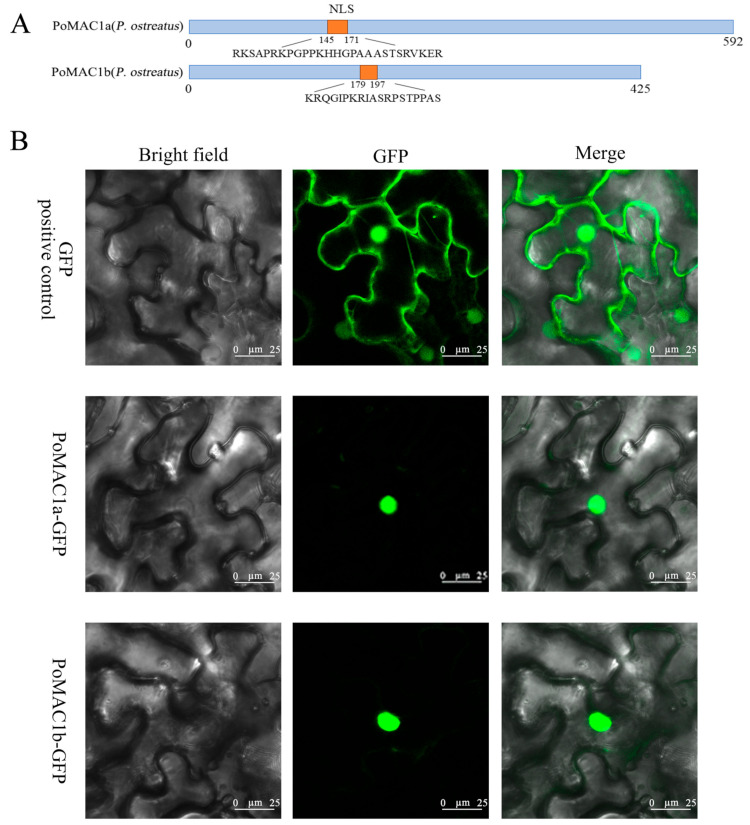
The subcellular localization of PoMAC1a and PoMAC1b. (**A**) Prediction of NLS for PoMAC1a and PoMAC1b. (**B**) The subcellular localization results of PoMAC1a and PoMAC1b. GFP (Green Fluorescent Protein), PoMAC1a GFP, and PoMAC1b GFP were expressed in tobacco leaves; bar = 25 μm.

**Figure 4 jof-10-00013-f004:**
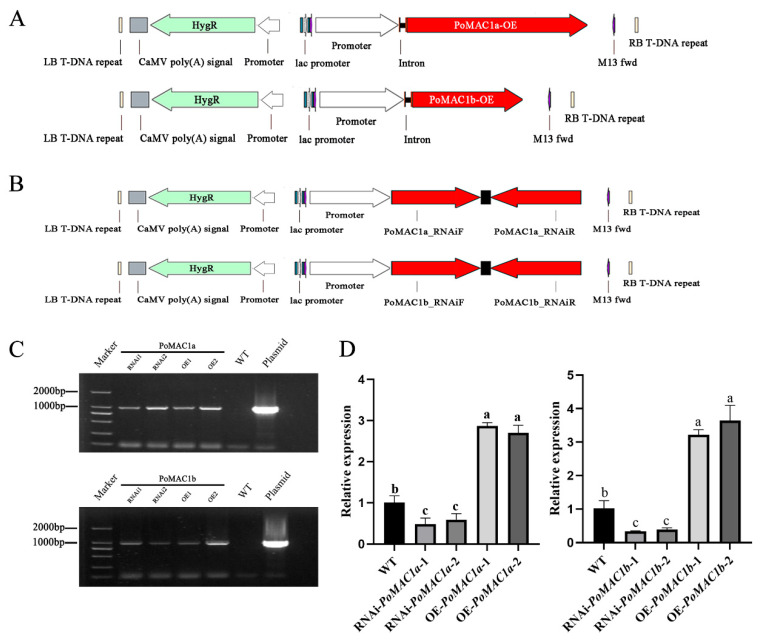
Verification of OE and RNAi strains. (**A**) Construction method of OE vectors. (**B**) Construction method of RNAi-silencing vectors. (**C**) PCR assay of *hyg* in *P. ostreatus* mutant strains, WT, and plasmid. (**D**) qPCR analysis of the relative expression of *PoMAC1* in the tested strains. Different letters indicate significant differences for the comparison of samples (*p* < 0.05 according to Duncan’s test).

**Figure 5 jof-10-00013-f005:**
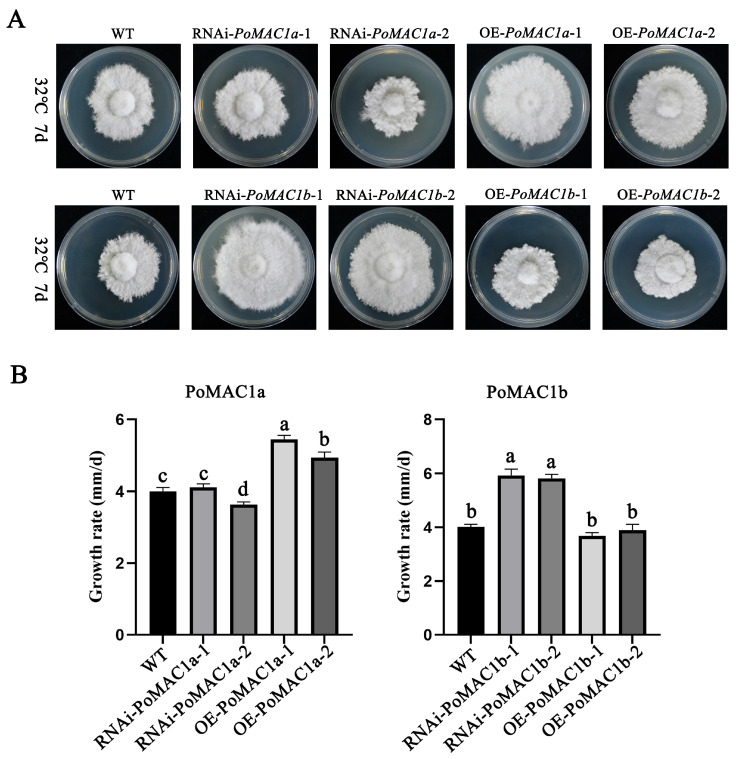
Detection of *PoMAC1* mutant strains’ tolerance to 32 °C heat stress. (**A**) Colony morphology of different strains at 32 °C. (**B**) The growth rate of different strains at 32 °C. Different letters indicate significant differences for the comparison of samples (*p* < 0.05 according to Duncan’s test).

**Figure 6 jof-10-00013-f006:**
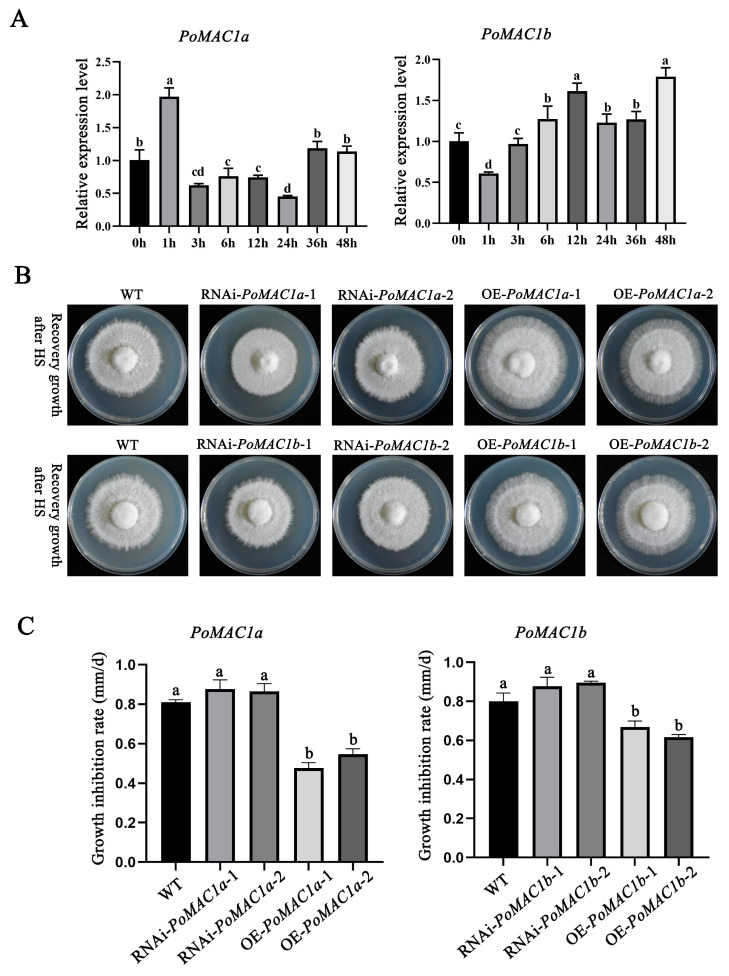
*PoMAC1* participates in the resistance of mycelia to 40 °C heat stress. (**A**) The expression levels of *PoMAC1a* and *PoMAC1b* under 40 °C stress at different times. (**B**) The effect of *PoMAC1a* mutant strains on the recovery growth rate of mycelia after heat stress. (**C**) The effect of *PoMAC1b* mutant strains on the recovery growth rate of mycelia after heat stress. The values are the mean ±SE of three independent experiments. Different letters indicate significant differences for the comparison of samples (*p* < 0.05 according to Duncan’s test).

**Figure 7 jof-10-00013-f007:**
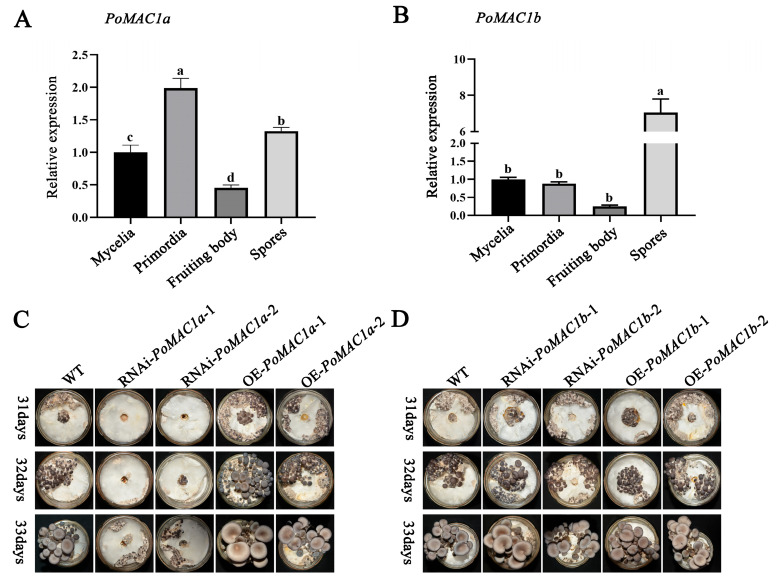
*PoMAC1a* promotes primordia formation in *P. ostreatus*. (**A**) The expression patterns of *PoMAC1a* during different developmental stages. (**B**). The expression patterns of *PoMAC1b* during different developmental stages. (**C**) Mushroom production of the WT strain and *PoMAC1*a mutant strains. (**D**) Mushroom production of the WT strain and *PoMAC1b* mutant strains. The values are the mean ± SE of three independent experiments. Different letters indicate significant differences for the comparison of samples (*p* < 0.05 according to Duncan’s test).

## Data Availability

The data of all results in this study are included in the manuscript and Appendix A. If necessary, the data can be obtained by contacting the corresponding author.

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
