# Peer review of "MAC Family Transcription Factors Enhance the Tolerance of Mycelia to Heat Stress and Promote the Primordial Formation Rate of Pleurotus ostreatus"

_jof, 2023, doi:10.3390/jof10010013_

Round 1

Reviewer 1 Report

Comments and Suggestions for Authors

The article submitted by Tqa et al. bioinformatically identifies two genes presumed to encode copper-sensitive transcription factors. The authors conduct an analysis of the sequences of both genes and compare them with other related MAC proteins, identifying key structural motifs. They verify their expression and function under various experimental conditions.

The article is well-written overall, and the results it presents are interesting, particularly given the relatively underdeveloped nature of this field in Pleurotus ostreatus. I believe it is a generally informative article in the field.

There are, however, some crucial points that the authors need to address or clarify during the article preparation process.

1. The results of the intracellular localization of the transcription factors are not clear. The presented images are of low quality, and it is not possible to identify cellular structures that would confirm their claim of nuclear localization. Additionally, the authors do not indicate nuclear localization signals in the protein sequences that would allow for a computational presumption of protein localization in the nucleus, potentially facilitating an appropriate experiment with the native protein. In any case, I believe that at this point, it is essential to enhance the quality of Figure 3 to ensure convincing results.

2.- The data regarding the response to H2O2 are questionable. The authors focus more on the stress tolerance capacity in the MACAaOE and MAC1b-i mutants than on demonstrating that MAC1a-i is more sensitive than the control. Regarding the MAC1b mutants, the images do not display appreciable differences. The authors should attempt to provide quantitative data because, from the images provided, the conclusion presented in the study cannot be deduced.

3.- Similarly, the results presented for growth and recovery from thermal stress do not appear clear. At first glance, there are no discernible differences in growth that support their conclusions. Once again, I believe it is necessary to quantify the results to make them convincing.

4.- In Figure 7D, I believe there is a mistake in the names of the mutants.

5.- Also, in Figures 7C and 7D, it would be advisable to explain why the control (WT) shows significantly more fruiting bodies in the panel corresponding to 7 days in Figure D than in Figure C.

Comments on the Quality of English Language

The use of English is accurate, and the work presented is easy to understand.

Reviewer 2 Report

Comments and Suggestions for Authors

The article Copper-sensing transcription factor PoMAC1 enhances the tolerance of mycelia to heat stress and promotes the primordial formation rate of Pleurotus ostreatus is very interesting, showing the biological role that the transcription factor has in one of the most cultivated fungi worldwide. Some comments are:

- I suggest improving the wording of section 2.6 since it does not indicate the growing conditions (humidity, temperature, light, etc.)
- In section 2.7, when they put the strains at 28ºC for recovery growth, they do not indicate the time left. Isn't it important to mention it?
- Review and correct the place of figures 1 and 2 and the legend
- The information in section 3.3 could be reinforced if the authors mentioned the characterization of the crop (crop cycle, biological efficiency, production rate, harvest period).

Author Response

Response to Reviewer 2 Comments

Manuscript ID: jof-2520573

Copper-sensing transcription factor PoMAC1 enhances the tolerance of mycelia to heat stress and promotes the primordial formation rate of Pleurotus ostreatus

Kexing Yan, Lifeng Guo, Benfeng Zhang, Mingchang Chang, Junlong Meng, Bing Deng, Jingyu Liu, and Ludan Hou

Dear Editor and Reviewer 2,

We are very grateful for your comments and suggestions for the manuscript. The sections revised according to the reviewer’s suggestions are highlighted as red text in the revised manuscript. The point-by-point replies to the reviewers’ comments are listed below:

Reviewer's Comments and Our Responses:

Dear Reviewer: 2

Thank you for the experts' comments on our manuscript. We listed the questions mentioned by Reviewer 2 in the manuscript one by one and answered them:

Point 1:  I suggest improving the wording of section 2.6 since it does not indicate the growing conditions (humidity, temperature, light, etc.)

Response: Thank you for your suggestion. We have added the growth conditions in section 2.6.(new lines 151-154)

Point 2: In section 2.7, when they put the strains at 28ºC for recovery growth, they do not indicate the time left. Isn't it important to mention it?

Response: Thank you for your question. The time to resume growth is very important. We have added the time to resume growth in section 2.7 (new lines 160-163)

Point 3: Review and correct the place of figures 1 and 2 and the legend

Response: Thank you for your suggestion. They were corrected.

Point 4: The information in section 3.3 could be reinforced if the authors mentioned the characterization of the crop (crop cycle, biological efficiency, production rate, harvest period).

Response: Thank you for your suggestion. According to your suggestion, we have included the impact of mutant strains on the growth and development cycle of fruiting bodies in the manuscript (new lines 311-315).

Reviewer 3 Report

Comments and Suggestions for Authors

Introduction

Introduction does not give any que about how the authors decided to work with PoMAC1a and PoMAC1b genes??? What is nutritional growth, vegetative growth is a better expression

Materials and Methods

The description of the identification and isolation of genes encoding  PoMAC1 and PoMAC1 and the origin of CCMSSC 00389 genome database is  very insufficient. In order to understand one has to go to Methods of publication “Expression patterns of two pal genes of Pleurotus ostreatus across developmental stages and under heat stress” by the authors (BMC Microbiology volume 19, Article number: 231, 2019). The same concerns the construction of PoMAC1 and PoMAC1 overexpression strains. These parts of methods must be improved.

The manuscript is a new one, although it follows closely the steps used to identify the P. ostreatus pal genes. The reader of the present manuscript does not want to go back to reference 13 to understand methods used in the research of P. ostreatus MAC genes.

Results

The structure and phylogenetic position of PoMAC1 and PoMAC1 are well reported except a few small mistakes:

Fig. 2B text The Cu-fist motif is grey;  Rep I red and Rep II blue

Fig. 3. Why GFP signal is present in the nucleus in the positive control?

Fig.7 text D is about PoMAClb

RNAi  of PoMAC1b-1 and PoMAC1b-2 increases the vegetative growth of P. ostreatus mycelium significantly. No explanation, neither in Discussion.

Discussion

Discussion concentrates in comparison of the present results with those obtained in yeasts and in Aspergillus species about the number of mac genes and rep motifs. The lines 311-316 should be written more clearly. Reader remains wondering why P. ostreatus mac genes have a role in heat and other stresses as well in the development of fruiting bodies. There is just a couple of rows at the end of Discussion, could be more detailed.

Comments on the Quality of English Language

No comments

Round 2

Reviewer 1 Report

Comments and Suggestions for Authors

Yan et al.'s responses to the comments from the initial review are satisfactory, and the article they are now submitting has been significantly improved compared to the initial version.

I want to provide some feedback to contribute to improving an article that I find interesting, and that provides relevant information.

In the title, the authors refer to PoMAC1 and its role in increasing tolerance to thermal stress and primordia formation. However, this is only true for PoMAC1a. I suggest considering a title change using "MAC family transcription factors" instead of explicitly mentioning PoMAC1.

Repeating keywords from the title in the Keywords section is not advisable as it does not enhance the article's searchability for computer systems. I would suggest that the authors use other keywords such as "Oyster mushroom," "gene silencing," "gene overexpression," "stress tolerance," and "fruiting" as alternatives.

On line 189, using " approximately " is inappropriate because the authors know the number of predicted amino acids in the sequence.

In line 205, use "Cu-fist" instead of "Cu-Fist" to maintain coherence in the text.

Lines 268-276 are somewhat chaotic. Assuming we consider an increase in growth rate and overall growth as positive, it is incorrect to state that PoMAC1a has a negative effect at this temperature since overexpression of this gene (in both experiments) leads to a higher growth rate and total growth. On the other hand, gene silencing does not seem to have effects at this temperature. Additionally, the overexpression of PoMAC1b has no effects, while silencing promotes growth. Therefore, the results suggest that PoMAC1a is a growth-promoting factor, while PoMAC1b is a growth-repressing factor at this temperature.

Indeed, what the authors define in lines 269 and 270 is incorrect since OE-PoMAC1b1 and OE-PoMAC1b2 do not show an increase in growth rate; instead, the silenced mutants exhibit this effect.

In Figure S2, we encounter a similar issue: while PoMAC1a is a factor for H2O2 tolerance, as its silencing reduces growth, PoMAC1b is a factor for sensitivity to this oxidizing agent, as its silencing results in increased growth in the mutant.

In Figure 6B, there is a fascinating detail that I assume the authors have observed: the growth zone of the overexpression mutants is visibly more prominent than that of the control (WT) and the silenced mutants. This is evident in the peripheral zone of the colony (approximately 0.5 cm), with a finer and exploratory mycelium showing recovery from thermal shock. This exploratory growth edge is not as clearly visible in the WT or the silenced mutants, which exhibit colonies with compact edges.

In line 278, I would refrain from using the word "trend" because what they observe is overexpression of PoMAC1a only in the first sample. The observed results are consistent with a response to the shock rather than an adaptation process, which suggests that using the word "trend" might not be appropriate.

In line 326, I would use "yeasts" in the plural form, and in line 327, I would use "filamentous fungi" or "basidiomycetes" instead of "large fungi."

Finally, I want to congratulate the authors again for improving the article.

Author Response

Response to Reviewer Comments

Manuscript ID: jof-2520573

MAC family transcription factors enhance the tolerance of mycelia to heat stress and promote the primordial formation rate of Pleurotus ostreatus

Kexing Yan, Lifeng Guo, Benfeng Zhang, Mingchang Chang, Junlong Meng, Bing Deng, Jingyu Liu, and Ludan Hou

We greatly appreciate your affirmation of the revised manuscript. Based on feedback from the reviewers, the revised content section is highlighted in red text in the revised manuscript. The response to the feedback from the reviewers is as follows:

Reviewer's feedback and Our Responses:

Point 1: In the title, the authors refer to PoMAC1 and its role in increasing tolerance to thermal stress and primordia formation. However, this is only true for PoMAC1a. I suggest considering a title change using "MAC family transcription factors" instead of explicitly mentioning PoMAC1.

Response: Thanks for your suggestion. We have revised the title of the article to "MAC family transfer factors enhance the tolerance of my family to heat stress and promote the primary formation rate of Pleurotus ostreatus".

Point 2: Repeating keywords from the title in the Keywords section is not advisable as it does not enhance the article's searchability for computer systems. I would suggest that the authors use other keywords such as "Oyster mushroom," "gene silencing," "gene overexpression," "stress tolerance," and "fruiting" as alternatives.

Response: Thanks for your suggestion. We have replaced the keywords with "Oyster Mushroom," "gene silencing," "gene overexpression," "stress tolerance," and "fruiting body". (new line 29)

Point 3: On line 189, using " approximately " is inappropriate because the authors know the number of predicted amino acids in the sequence.

Response: Thanks for your correction. It was removed. (new lines 187-189)

Point 4: In line 205, use "Cu-fist" instead of "Cu-Fist" to maintain coherence in the text.

Response: Thanks for your suggestion. It was corrected. (new line 204)

Point 5: Lines 268-276 are somewhat chaotic. Assuming we consider an increase in growth rate and overall growth as positive, it is incorrect to state that PoMAC1a has a negative effect at this temperature since overexpression of this gene (in both experiments) leads to a higher growth rate and total growth. On the other hand, gene silencing does not seem to have effects at this temperature. Additionally, the overexpression of PoMAC1b has no effects, while silencing promotes growth. Therefore, the results suggest that PoMAC1a is a growth-promoting factor, while PoMAC1b is a growth-repressing factor at this temperature. Indeed, what the authors define in lines 269 and 270 is incorrect since OE-PoMAC1b1 and OE-PoMAC1b2 do not show an increase in growth rate; instead, the silenced mutants exhibit this effect.

Response: Thank you for your correction. We have corrected the incorrect description in the text. (new lines 267,270)

Point 6: In Figure S2, we encounter a similar issue: while PoMAC1a is a factor for H2O2 tolerance, as its silencing reduces growth, PoMAC1b is a factor for sensitivity to this oxidizing agent, as its silencing results in increased growth in the mutant.

Response: Thank you for your suggestion. We have revised the results of Figure S2 (new lines 271-273)

Point 7: In Figure 6B, there is a fascinating detail that I assume the authors have observed: the growth zone of the overexpression mutants is visibly more prominent than that of the control (WT) and the silenced mutants. This is evident in the peripheral zone of the colony (approximately 0.5 cm), with a finer and exploratory mycelium showing recovery from thermal shock. This exploratory growth edge is not as clearly visible in the WT or the silenced mutants, which exhibit colonies with compact edges.

Response: Thank you for your suggestion. We will provide a more detailed description of the results in Figure 6B. (new lines 282-286)

Point 8: In line 278, I would refrain from using the word "trend" because what they observe is overexpression of PoMAC1a only in the first sample. The observed results are consistent with a response to the shock rather than an adaptation process, which suggests that using the word "trend" might not be appropriate.

Response: Thank you for your suggestion. We have rephrased the results in the article. (new lines 279-282)

Point 9: In line 326, I would use "yeasts" in the plural form, and in line 327, I would use "filamentous fungi" or "basidiomycetes" instead of "large fungi."

Response: Thank you for your suggestion. We have made the corresponding modifications. (new lines 330,331)